# Antimicrobial Efficacy of Five Wound Irrigation Solutions in the Biofilm Microenvironment In Vitro and Ex Vivo

**DOI:** 10.3390/antibiotics14010025

**Published:** 2025-01-03

**Authors:** Anja L. Honegger, Tiziano A. Schweizer, Yvonne Achermann, Philipp P. Bosshard

**Affiliations:** 1Department of Dermatology, University Hospital Zurich, University of Zurich, 8091 Zurich, Switzerland; anjalinn.honegger@uzh.ch (A.L.H.); philipp.bosshard@usz.ch (P.P.B.); 2Department of Cranio-Maxillo-Facial and Oral Surgery, University Hospital Zurich, University of Zurich, 8091 Zurich, Switzerland; 3Internal Medicine, Hospital Zollikerberg, 8125 Zollikerberg, Switzerland

**Keywords:** PJI, biofilm, irrigation, wound irrigation solution

## Abstract

**Background/Objectives:** Periprosthetic joint infections (PJI) are difficult to treat due to biofilm formation on implant surfaces and the surrounding tissue, often requiring removal or exchange of prostheses along with long-lasting antibiotic treatment. Antiseptic irrigation during revision surgery might decrease bacterial biofilm load and thereby improve treatment success. This in vitro study investigated and compared the effect of five advanced wound irrigation solutions to reduce bacterial burden in biofilm microenvironment. **Methods:** We treated in vitro biofilms grown on titanium alloy implant discs with clinical bacterial strains isolated from patients with PJIs, as well as abscess communities in a plasma-supplemented collagen matrix. The biofilms were exposed for 1 min to the following wound irrigation solutions: Preventia^®^, Prontosan^®^, Granudacyn^®^, ActiMaris^®^ forte (‘Actimaris’), and Octenilin^®^. We measured the bacterial reduction of these irrigation solutions compared to Ringer–Lactate and to the strong bactericidal but not approved Betaseptic solution. Additionally, ex vivo free-floating bacteria isolated directly from clinical sonication fluids were treated in the same way, and regrowth or lack of regrowth was recorded as the outcome. **Results:** Irrigation solutions demonstrated variable efficacy. The mean CFU log_10_ reduction was as follows: Octenilin, 3.07, Preventia, 1.17, Actimaris, 1.11, Prontosan, 1.03, and Granudacyn, 0.61. For SACs, the reduction was: Actimaris, 8.27, Octenilin, 0.58, Prontosan, 0.56, Preventia, 0.35, and Granudacyn, 0.24. **Conclusions:** All solutions achieved complete bacterial eradication in all tested ex vivo sonication fluids, except Granudacyn, which was ineffective in 33% of the samples (2 out of 6). Advanced wound irrigation solutions have the potential to reduce bacterial burden in the biofilm microenvironment. However, their efficacy varies depending on bacterial species, growth state, and the composition of the irrigation solution. While Octenilin should be avoided for deep tissue irrigation due to its potential to cause tissue necrosis, the clinical benefit of wound irrigation solutions in infection prevention warrants further investigation in prospective clinical trials.

## 1. Introduction

The use of orthopedic implants, particularly prosthetic joints, has steadily increased over recent decades, which is due to the increasing prevalence of osteoarthritis and the resulting need for prostheses in aging populations [1,2]. Despite advancements in surgical techniques and infection prevention strategies, periprosthetic joint infections (PJIs) continue to be one of the most serious complications following these procedures. Occurring in approximately 1–2% of all hip and knee arthroplasties, PJIs are associated with increased morbidity, mortality, prolonged hospital stays, and substantial healthcare costs [3,4]. Current treatment protocols fail to achieve successful outcomes in >15% of cases [5,6,7]. Treating PJIs is particularly difficult due to the ability of bacteria to form biofilms on the surface of orthopedic implants and in surrounding tissues [8,9,10]. These biofilms act as a protective barrier, thereby complicating eradication efforts. Therefore, surgical treatment of PJI nowadays commonly includes the use of topically active surgical irrigation solutions, aimed at reducing the bacterial load in the PJI microenvironment.

Biofilms consist of bacterial communities embedded in a self-produced extracellular matrix that shields the bacteria and alters their growth behavior, thereby preventing the penetration of antibiotics and hindering the immune system’s ability to clear the infection. Biofilm matrices can be made up of various components, such as proteins, lipids, and sugars as well as DNA, depending on the specific bacterium [8]. These different components might also confer resistance toward certain substances [11,12,13]. Within a biofilm, bacteria are more tolerant to antibiotics and evade host immune responses, which makes PJIs more difficult to treat compared to other soft tissue infections [14]. As a result, managing PJIs often requires surgical intervention combined with prolonged antibiotic therapy [15,16]. The choice of treatment—whether debridement, antibiotics, and implant retention (DAIR), or one- or two-stage exchange arthroplasty—depends on the timing of infection, pathogen virulence, and patient factors. While the outcomes of PJI treatment can vary, achieving infection control often requires a combination of meticulous surgical debridement and prolonged antibiotic therapy [17].

A key component of surgical management is the irrigation and debridement of the infected area, aimed at removing residual biofilm bacteria and non-viable tissue [18]. Since it is known that as few as 100 colony-forming units (CFUs) of bacteria can cause implant-associated infection, reducing the bacterial burden in the PJI microenvironment is a crucial factor for preventing re-infection [19]. While normal saline has traditionally been used for irrigation during septic surgery, recent interest has grown in additional treatments such as irrigation solutions, including surfactants, antiseptics, and antibiotics, to enhance debridement and bacterial reduction [20,21]. However, antiseptic irrigation solutions used during arthroplasty surgery are primarily intended to prevent postoperative infections by eliminating any remaining bacteria at the end of the procedure, rather than as part of a septic surgery. In addition, despite the increasing number of available irrigation solutions, there is still no consensus on the optimal type and protocol for managing acute or chronic PJIs. Current guidelines from WHO and CDC [22] offer limited guidance, and high-quality preclinical and clinical data comparing the efficacy of different solutions are scarce. Additionally, the specific components of the infectious microenvironment are often not considered in experimental studies. This knowledge gap is a major challenge, as selecting the most effective irrigation solution could significantly impact treatment outcomes, especially in complex infections like PJIs. Several studies have investigated the efficacy of single irrigation solutions, such as chlorhexidine [23,24,25,26,27] and Betadine (Povidone iodine [3.24%]) [24,28,29], as well as Bactisure (EtOH and Benzalkonium chloride) [19,20]. However, both chlorhexidine and non-diluted Betadine are not indicated for intraoperative irrigation due to concerns about toxicity, iodine allergies, and thyroid dysfunction [30]. Therefore, we chose to compare the efficacy of various novel non-alcohol-based wound irrigation solutions in reducing bacterial burden in the biofilm microenvironment in relevant laboratory models in order to explore a potential additional use beyond wound treatment. This was done in comparison to standard Ringer’s lactate solution, which has no antibacterial effect, and the off-label use of Betaseptic—an alcohol-based solution with strong bactericidal activity—which served as a positive control. The selection of these irrigation solutions was based on existing experimental evidence regarding their effectiveness against planktonic bacteria [30,31,32,33,34].

## 2. Results

### 2.1. Biofilms on Titanium Alloy (TAV) Discs In Vitro Assay

Biofilm formation on the implant itself is considered the major culprit for PJI progression, due to impaired host immune activity in the presence of foreign material [19]. Therefore, as the main experimental model, we chose to use a biofilm on implant material in vitro model. We compared the efficacy of the five different irrigation solutions in reducing the absolute CFUs/mL of *S. aureus*, *S. epidermidis*, *E. coli*, and *C. acnes* biofilms on titanium discs. The respective active compounds and solvents of each irrigation solution are listed in Table 1. Based on their various active components, the solutions are proposed to act widely diverse, as indicated by the corresponding companies. Preventia and Prontosan present combinations of surfactants (degrading biofilm matrix and detaching bacteria from surfaces) and the antimicrobial compound polyhexanide (directly eradicating bacteria). Granudacyn uses hypochlorous acid to attack the cell wall and increase permeability thus causing cell rupture by osmolysis. Actimaris employs a combination of sea salt and sodium hypochlorite for antimicrobial effect. Octenilin uses, as the main active component, Octenidine HCl for antimicrobial and biofilm action. Ringer–Lactate has no direct antimicrobial effect but is rather employed to physically remove bacteria in the PJI microenvironment. Finally, the skin antiseptic Betaseptic uses a combination of EtOH and PVP-Iodine. When testing the solutions on *S. aureus* biofilms, Octenilin was the only irrigation solution that showed a statistically significant reduction of *S. aureus* by 2.06 log_10_ (Figure 1A, *S. aureus*). Although Preventia, Prontosan, Granudacyn, and Actimaris reduced bacterial counts to some extent, the effect was not significant. For *S. epidermidis*, none of the tested irrigation solutions achieved a significant reduction in biofilm reduction (all < 1 log_10_). The absolute log_10_ CFUs/mL were all in a similar range to the Ringer’s lactate control (Figure 1A, *S. epidermidis*). For *E. coli*, Octenilin again demonstrated a statistically significant biofilm reduction of 3.88 log_10_ (Figure 1A, *E. coli*) while all other solutions showed some reduction, but no significant effect. Notably, Betaseptic, that is not indicated for wound irrigation but for skin disinfection, achieved complete eradication of *S. aureus*, *S. epidermidis*, and *C. acnes*, but failed to completely eradicate *E. coli*. Finally, for *C. acnes*, Octenilin was again the most effective, achieving a statistically significant 5.94 log_10_ reduction (Figure 1A, *C. acnes*). Although not statistically significant, Prontosan achieved a >2 log_10_ reduction, while Preventia and Actimaris showed >1.5 log_10_ reduction, and Granudacyn showed >0.5 log_10_ reduction. Overall, the *C. acnes* isolate was the most sensitive of the tested isolates, while *S. epidermidis* showed the least reduction in CFUs/mL (Figure 1B). Among the five tested solutions, Octenilin was the most effective (Figure 1C). The dataset to Figure 1 is available as Appendix A.

### 2.2. Staphylococcal Abscess Communities (SACs) in Collagen Tissue In Vitro Assay

In a second step, we assessed and compared the efficacy of the irrigation solutions in reducing the bacterial load within SACs grown in collagen, representing bacterial colonization of the periprosthetic tissue within 24 h (Figure 2A). SACs have been shown to readily occur during bone and joint infections and can affect either soft tissue or bone [35]. They have been shown to confer significantly elevated tolerance towards antibiotics and protection against the host immune system [36]. They are believed to be one of the major factors involved in infection persistence and recurrence [37,38]. When treating these SACs with the irrigation solutions, we observed that Actimaris was the only solution that reached a statistically significant reduction of 8.27 log_10_, resulting in complete eradication (Figure 2B). Actimaris even outperformed Betaseptic, which achieved a reduction of 7.2 log_10_. None of the other irrigation solutions resulted in a reduction greater than 0.6 log_10_. The dataset to Figure 2 is available as Appendix A.

### 2.3. PJI Sonication Solutions Ex Vivo Assay

Finally, we determined and compared the efficacy of the irrigation solutions in eradicating planktonic bacteria seeding-form biofilms formed on implants. Bacterial detachment from biofilms is a key step in the biofilm life cycle to ensure the survival of bacteria [8]. By detachment of microcolonies or single planktonic cells, bacteria can seed to new locations and establish novel biofilms there, thereby increasing their chances of surviving in the given environment. In order to adequately reflect this scenario, we chose to use clinical samples containing planktonic bacteria originating from orthopedic implants that were detached by sonication. We used six sonication fluids from implants with confirmed infections with *S. aureus*, *S. epidermidis*, or *Enterobacter cloacae*. Since the bacterial load of sonication fluids is known to be low (Table 2), we chose to first concentrate them and then challenge concentrated aliquots with the irrigation solutions for 1 min. Due to the low bacterial load, we decided to use bacterial growth yes/no as the readout. All solutions except Granudacyn (which showed 66% efficacy) achieved complete eradication (i.e., no growth) (Table 2). Granudacyn failed to eradicate two samples containing *S. aureus*.

## 3. Discussion

Improved surgical treatment of PJI may increase the chances of successful treatment outcomes and thereby decrease morbidity and healthcare costs. Since it is known that wound irrigation solutions exhibit rapid bactericidal activity against planktonic bacteria, we investigated whether these commercially available solutions were able to reduce the bacterial load in the PJI biofilm microenvironment in vitro. Using different models that adequately reflect the PJI microenvironment consisting of biofilms on implants or embedded in tissue as well as planktonic bacteria originating from biofilms, we observed that the efficacy of irrigation solutions depends on the bacterium and the growth phenotype (biofilm or planktonic) as well as the bacterial localization in an abscess. To the best of our knowledge, no study has yet comparatively assessed the efficacy of various novel non-alcohol-based wound irrigation solutions in reducing bacterial burden in the PJI microenvironment.

Among the tested wound irrigation solutions, Octenilin demonstrated the overall highest efficacy and was the only solution to show a statistically significant reduction of bacteria within biofilms for *S. aureus*, *E. coli*, and *C. acnes*. The effect against *E. coli* and *C. acnes* was even bactericidal, i.e., a log_10_ reduction >3 was achieved, meaning that >99.9% of the total number of bacteria have been killed [39]. This is rather remarkable since biofilms, compared to planktonic bacteria, are more difficult to eradicate. Octenilin has Octenidine HCl as its main active substance, which is known to have a microbicidal effect for planktonic *S. aureus* in just 1 min [32,33]. Another study has shown that Octenidine HCl prevents and disintegrates biofilms of *S. aureus* on different materials independent of the presence or absence of serum proteins [40]. This study also demonstrated the accumulation of dead cells within the biofilm as well as loss in biofilm architecture. The excellent efficacy against *S. aureus* biofilms, but to a lesser extent against *Pseudomonas aeruginosa*, was further confirmed [12]. Based on these combined findings, one could postulate that the biofilm matrix composition might influence the efficacy of Octenidine HCl to penetrate into the biofilm. Rzycki et al. demonstrated that Octenidine HCl presents electrostatic selectivity towards charged lipids [11], which can be found in the biofilm matrix of *S. aureus* [41], but potentially not in biofilms formed by other species. However, it has to be noted that Octenilin is not indicated for intraoperative irrigation due to its potential to cause aseptic tissue necrosis due to the disruption of cell membrane integrity of mammalian cells when retained in tissue.

Interestingly, no solution was able to significantly reduce bacterial burden in *S. epidermidis* biofilms. Since *S. epidermidis* can be considered to be a lesser virulent version of *S. aureus*, the species excels at forming biofilms highly tolerant to withstand antimicrobial treatment and the immune response [42,43], which could explain our observation. *C. acnes* showed the highest susceptibility towards the irrigation solutions, followed by *E. coli* and *S. aureus*. While *S. aureus* and *E. coli* are fast-growing bacteria and hence rapid biofilm formers, *C. acnes* with its slow metabolism also requires more time to form mature biofilms [44]. However, since *S. epidermidis*, at least compared to *S. aureus* and *E. coli*, is also a slow-growing bacterium, a further reason for the observed differences could lie in the composition of the biofilm matrices. While the matrices of *C. acnes* as well as *E. coli* biofilms (the two most susceptible strains to the irrigation solutions) consist mostly of exopolysaccharides followed by proteins, DNA, and lipids [45,46], the *S. aureus* biofilm matrix is mostly made up of proteins and extracellular DNA [47,48,49], and the *S. epidermidis* biofilm matrix encompasses poly-N-acetylglucosamine as the main component [50]. However, whether the biofilm matrix components, especially poly-N-acetylglucosamine, play an important role in tolerance against surgical irrigation solutions remains to be investigated. Of note, despite the most significant efficacy observed with Betaseptic across all tested bacteria, Betaseptic is a skin antiseptic and not indicated for intraoperative use, also due to potential tissue toxicity issues [30].

When assessing the efficacy of the wound irrigation solutions against SACs, Actimaris was the only wound irrigation solution to demonstrate significant activity against a small abscess (SACs). Actimaris was even superior to Betaseptic. In addition to the role of the implant itself, the periprosthetic tissue is an important niche for bacterial persistence [51,52]. Bacterial colonization of the tissue can either occur via infection of and persistence within host cells [53] or in the form of tissue-embedded biofilms with *S. aureus*, i.e., abscesses [54]. In order to mimic this particular situation, we employed the previously published SAC model [36,55]. The SAC model is an in vitro representation of macroscopic abscesses surrounded by a fibrin capsule formed within a plasma-supplemented collagen type I matrix. Except for Actimaris, which completely eliminated the SACs and was even superior to Betaseptic, none of the other solutions achieved any remarkable reduction. As previously demonstrated, the fibrin capsule around the SACs confers increased tolerance towards antibiotics and neutrophils [36], which could potentially explain the negligible effect of the irrigation solutions. Since Actimaris reduces fibrin depositions in wounds and thereby accelerates wound healing [56], one explanation for the observed strong effect of Actimaris in the SAC model would be the degradation of the SACs fibrin capsule and hence direct contact with susceptible bacteria. Notably, abscesses are characterized by an acidic pH, which is important for their formation and stability [57,58]. Since Actimaris is alkaline, it could potentially cause rapid disintegration of the SACs in the collagen tissue making the bacteria vulnerable.

When assessing planktonic bacteria originating from clinical biofilms on implants, apart from Granudacyn, all tested wound irrigation solutions led to complete eradication. Attacking seeding bacteria from the PJI microenvironment is usually overlooked in experimental PJI models, yet it plays a crucial role during revision surgery. Bacteria detaching from sites of infections and becoming planktonic were generally considered to become susceptible to treatments and the immune system [59]. However, it was then shown that bacteria released from biofilms were more virulent [60], more effective at infecting immune cells [61], and potentially still show antibiotic tolerance [13], potentially due to reduced metabolic activity [62]. Assuming these liberated bacteria would still freely roam in the PJI microenvironment, even as few as 100 CFUs could then cause re-infection of the newly inserted implant [19]. Therefore, in order to best represent these biofilm-originating bacteria, we used sonication solutions obtained from sonicated prostheses with confirmed PJI. Of note, we generally only obtained samples relatively low in bacterial numbers (~10^3^ CFUs total), which impeded us from testing the irrigation solutions on samples with elevated bacterial loss. However, considering the bacterial load from published studies, a total bacterial load of ~10^3^ CFUs per implant seems to be average [63]. Except for Granudacyn, all irrigation solutions led to complete bacterial eradication in the treated samples. Granudacyn, which failed to completely eradicate two samples containing *S. aureus*, has previously been suggested to be partially inhibited by wound exudate [31]. Therefore, the remaining wound exudate in these two samples might have limited the activity of Granudacyn.

This study also has some limitations. We only employed a contact time of 1 min for each irrigation solution. Some irrigation solutions are indicated to be left in contact for a prolonged time, up to 15 min. Hence, before using any of these wound irrigation solutions for PJI treatment in the clinical setting, the manufacturers instruction for use has to be considered carefully. However, we were aiming to identify irrigation solutions with rapid efficacy, since time is an important risk factor for surgical treatment outcome [64]. Further studies are needed to compare the antibiofilm efficacy in regard to different contact times. Furthermore, we employed only a static biofilm model with optimal conditions for bacteria to form highly mature biofilms. These might not reflect clinical biofilm maturity and matrix composition. Therefore, certain wound irrigation solutions might not show activity in our experimental set-up, while they nevertheless might show some effect in a real clinical scenario. We are aware that the selected parameters and conditions cannot reflect the clinical situation exactly, but they can help to better understand the effect of wound irrigation solutions in the PJI microenvironment.

Despite the importance of the antimicrobial efficacy profile of such irrigation solutions, another key factor is whether they show toxicity toward human cells and tissue. Though this was not within the scope of our study, further research on selected candidates should include toxicity testing and material compatibility studies. Based on our data suggesting that the efficacy of the irrigation solutions is linked to bacterial phenotype and location, combinatorial and/or sequential application of selected irrigation solutions could potentially enhance the antimicrobial efficacy. However, they may also inhibit each other, so a careful evaluation of all involved factors will be necessary for future studies exploring this approach. Finally, while all irrigation solutions showed some efficacy to reduce bacterial loads, this might already be sufficient when combined with antibiotics, since it was shown that the combination of a biofilm-targeting approach and antibiotics reduces the concentration and time of antibiotics required to eliminate biofilm bacteria [59]. This remains a hypothesis for future work.

## 4. Materials and Methods

### 4.1. Bacterial Strains and Cultivation

The following clinical isolates from patients with PJI were used for the biofilm assays: *Staphylococcus aureus* (BCI175), *Staphylococcus epidermidis* (BCI195), *Escherichia coli* (BCI0413), and *Cutibacterium acnes* (Y105). The bacterial biobank was approved by the institutional review board in Zurich, Switzerland (KEK Nr 2016-00145, KEK Nr 2017-01458). Fresh cultures were prepared from frozen cultures and were put on Brain Heart Infusion (BHI, Becton Dickinson, Heidelberg, Germany) agar plates for 24 h under aerobic conditions with 37 °C (*S. aureus*, *S. epidermidis*, and *E. coli*) or 3 days under anaerobic conditions for *C. acnes* (GENbags, bioMérieux, Mary-l’Etoile, France). Liquid cultures were made with colonies from the agar plates in (BHI broth in a 37 °C incubator with shaking overnight under aerobic conditions for *S. aureus, S. epidermidis*, and *E. coli* and shaking for three days under anaerobic conditions for *C. acnes*. Then, 1:1000 dilutions were made in fresh BHI medium. Samples were vortexed and used as start inoculum for biofilm assays right away.

### 4.2. Biofilm Assays

Biofilm assays were established as described previously [65]. In brief, sterilized orthopedic titanium alloy Ti-6A1-4V (TAV) discs (round, 10 mm in diameter and 1 mm in thickness) (provided by Synthes GmbH, Zuchwil, Switzerland) in 24-well plates served as attachment surfaces where *S. aureus*, *S. epidermidis*, *E. coli*, and *C. acnes* would grow their static biofilms. From the starting inoculum, 1 mL was put on each disc. The 24-well plates were incubated at 37 °C for 3 days (*S. aureus*, *S. epidermidis*, *E. coli*) or 4 days (*C. acnes*) before the BHI medium was exchanged with 1 mL fresh medium to ensure optimal growth of bacteria. After 6 days (*S. aureus*, *S. epidermidis*, and *E. coli*) or 8 days (*C. acnes*) of incubation, the BHI medium was removed, discs were transferred into new 24-well plates and washed two times with 1 mL phosphate-buffered saline (PBS, Gibco, ThermoFisher Scientific, Waltham, MA, USA) to remove any planktonic bacteria before treating discs with 1 mL of irrigation solutions.

### 4.3. Treatment with Irrigation Solutions

The five wound irrigation solutions used in these experiments were Preventia^®^ (Hartmann), Prontosan^®^ (B. Braun), Granudacyn^®^ (Mölnlycke), ActiMaris^®^ forte (‘Actimaris’), Octenilin^®^ (Schülke & Mayr), and the skin antiseptic Betaseptic (Mundipharma) (Table 1). We used Ringer’s lactate solution (B. Braun) as a negative control. The biofilm-titanium discs were treated with 1 mL of irrigation or control solution for 1 min. Then, the solutions were removed, and the discs were washed twice with 1 mL PBS. Again, 1 mL PBS was applied to the discs, and the 24-well plate was then sonicated in a water bath for 1 min at 40 Hz. Next, the sonication solutions were ten-fold diluted and drop-plated on BHI agar plates. Agar plates were incubated at 37 °C overnight (*S. aureus*, *S. epidermidis*, and *E. coli*) or anaerobically for 3 days (*C. acnes*) to enumerate CFUs/mL.

### 4.4. S. aureus Abscess Communities (SACs)

SACs were established as previously described with minor modifications [22,23]. A 0.5 mg/mL collagen type 1 solution (Rat Tail, Sigma Aldrich, St. Louis, MO, USA) was prepared from a 3 mg/mL stock by diluting 1:6 with fetal bovine serum (FBS, Gibco, ThermoFisher Scientific, Waltham, MA, USA) and 200 µL of it were put in a 24-well plate and incubated for 30 min at 37 °C. Once the gel polymerized, it was infected with the clinical *S. aureus* (BCI175) strain as follows: colonies from BHI plates were dissolved in PBS to an optical density (OD) of 0.5. Then, a 1:10^6^ dilution in FBS was made and 20 µL was pipetted onto each gel in the 24-well plate. After 30 min in the incubator at 37 °C, 500 µL FBS was put on top of the gel and SACs were grown overnight at 37 °C. The next day, SACs were treated with 1 mL of solutions for 1 min. Next, they were mixed thoroughly and 200 µL were collected to perform a 10-fold dilution series. Dilution series were drop-plated onto BHI agar plates as described above.

### 4.5. Sonication Fluids Treatment

Anonymized sonication fluids from already treated arthroplasty implants with a PJI were obtained from the Institute of Medical Microbiology at the University of Zurich. Bacteriological analysis by MALDI-TOF was also performed by the Institute of Medical Microbiology as part of their routine diagnostic procedure. At first, sonication fluids were centrifuged at 3000× *g* for 5 min. Then, the supernatants were removed and 100 µL Ringer’s lactate solution was used to resuspend the pellet. After vortexing, 10 µL of each sonication fluid was put in a 96-well plate. The sonication fluids were then treated with 190 µL of irrigation solutions for 1 min and then centrifuged again with 3000× *g*. The remaining pellet was washed with 200 µL PBS and resolved in 200 µL BHI. The 96-well plate was then placed in the 37 °C incubator overnight. As a readout, growth/no growth was selected, due to the low bacterial CFU load in the original sonication fluids.

### 4.6. Statistical Analysis

Statistical analysis was done using GraphPad Prism 10 (GraphPad, San Diego, CA, USA) with the employed statistical tests indicated in the respective figure legends.

## 5. Conclusions

To conclude, the aim of this study was to compare commercially available surgical irrigation solutions for their efficacy in reducing bacterial burden in the PJI microenvironment. We showed that the irrigation solutions all show at least some efficacy in reducing bacterial loads, but this efficacy depends on the (I) bacterium, (II) bacterial growth phenotype, i.e., planktonic or biofilm, and (III) whether the biofilm is formed on the surface of the material and thereby easily accessible by the irrigation solutions or within tissue and thereby less accessible. Furthermore, given the limited impact of the irrigation solutions on mature biofilms, they might only be used in adjunction to extensive surgical debridement, in order to eliminate potentially remaining microorganisms. This highlights the importance of considering the various factors composing the infectious PJI microenvironment for the future development of irrigation solutions aimed specifically at supporting the surgical treatment of PJI.

## Figures and Tables

**Figure 1 antibiotics-14-00025-f001:**
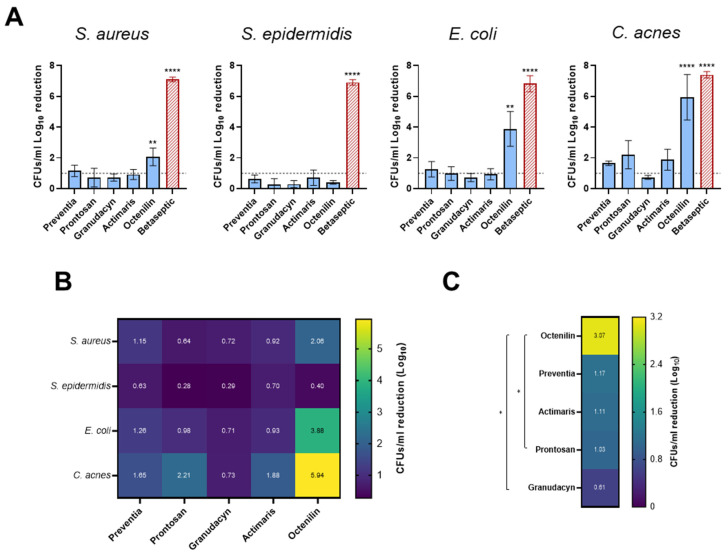
Antimicrobial efficacy of the irrigation solutions on biofilms formed on titanium alloy (TAV) discs. (**A**) Log_10_ CFUs/mL reduction of the irrigation solutions on 6-day-old *S. aureus*, *S. epidermidis*, and *E. coli*, as well as 8-day-old *C. acnes* biofilms formed on TAV discs. Results are depicted as a relative reduction to Ringer’s lactate solution (negative control) and are the mean (±SEM) of three independent experiments performed in duplicates. Betaseptic (red) is not a wound irrigation solution but was used as a positive control with strong bactericidal activity. (**B**) Heat map indicating the relative log_10_ CFUs/mL reductions across all irrigation solutions and bacterial species. (**C**) Heat map indicating the mean log_10_ CFUs/mL reduction of each irrigation solution across all bacteria. Statistical analyses in A and C were performed by two-way ANOVA. Statistically significant reductions are indicated with asterisks: *, *p* < 0.5; **, *p* < 0.1; ****, *p* < 0.0001.

**Figure 2 antibiotics-14-00025-f002:**
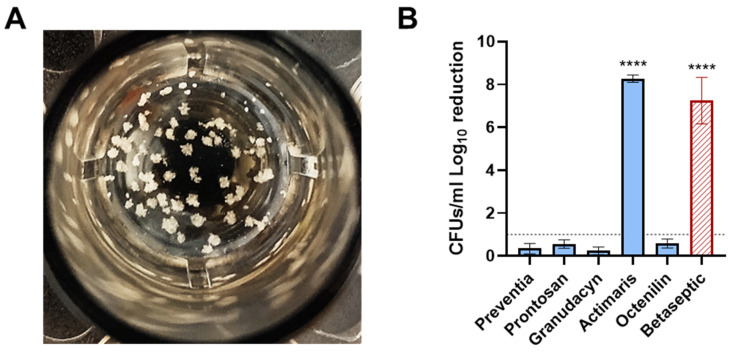
Antimicrobial efficacy of the irrigation solutions in the *S. aureus* abscess communities (SAC) model. (**A**) Representative image of SACs grown in collagen. (**B**) Log_10_ CFUs/mL reduction of the irrigation solutions on 24 h old SACs formed in plasma-supplemented collagen matrices. Results are depicted as a relative reduction to Ringer’s lactate solution (negative control) and are the mean (±SEM) of three independent experiments performed in duplicates. Betaseptic (red) is not a wound irrigation solution. Statistically significant reductions are indicated with asterisks: ****, *p* < 0.0001.

**Table 1 antibiotics-14-00025-t001:** Compositions of the five non-alcohol-based irrigation solutions.

Irrigation Solutions	Company	Active Ingredients	Solvent	Indication by the Company (Package Insert)
Preventia^®^	Hartmann (Heidenheim an der Brenz, Germany)	Polyhexanide (0.1%), poloxamer	H_2_O	- Conventional irrigation solution during surgery - Prevention of postoperative wound infection - Revisions (infectious or noninfectious due to loosening) or primary arthroplasties of knee and hip joints - Wound closure phase for all surgeries - Not indicated for use on cartilage
Prontosan^®^	B. Braun (Sempach, Switzerland)	Polyhexanide (0.1%), betaine (0.1%)	H_2_O	- Cleaning, irrigation, and moisturizing of acute, chronic, and infected skin wounds, and first- and second-degree burns - Intraoperative cleaning and rinsing of wounds - Loosening encrusted wound compresses and bandages
Granudacyn^®^	Mölnlycke (Schlieren, Switzerland)	Sodium chloride, hypochlorous acid (<0.005%), sodium hypochlorite (<0.005%)	H_2_O	- Cleaning and moisturizing of chronic, acute, surgical, and contaminated wounds as well as first- and second-degree burns
ActiMaris^®^ forte	ActiMaris (Appenzell, Switzerland)	Oxychlorit NaOCl (0.2%), sea salt (3%)	H_2_O	- Cleaning, humidification, decontamination, biofilm resolution, reduction of swelling, and physiological debridement of the following wounds: - Acute mechanical wounds - Postoperative wounds - Chronic wounds - Necrotic, malodorous wounds and ulcerating tumors, even with cavities (hollow spaces), venous ulcers - Thermal and chemical wounds (first- to third-degree burns) - Entry portals of urological catheters and PEG tubes and drains - Moistening and loosening of encrusted bandages and wound dressings - Local treatment of the skin and mucous membrane in inflammatory and infectious processes - Prevention of infection of the mucous membrane and skin, burns, and other wound types
Octenilin^®^	Schülke & Mayr (Frauenfeld, Switzerland)	Ethylhexylglycerol, octenidine HCl	H_2_O	- Moistening, cleansing, and removal of wound crusts consisting of necrotic tissue, pathogens, and biofilms - Removal of difficult-to-remove, caked dressings/wound pads

Betaseptic (Mundipharma Medical Company, Basel, Switzerland) with povidone–iodine (32.4 mg) and isopropanol (389 mg), ethanol 96% (389 mg) was used as a non-used alcohol-based solution with a strong bactericidal activity as a positive control.

**Table 2 antibiotics-14-00025-t002:** Efficacy of the irrigation solutions to eliminate planktonic bacteria in sonication fluids (biofilm-originating) compared to Betaseptic.

		Growth: Yes, 1; No, 0						
Pathogen	Concentration (CFUs/mL)	Ringer	Preventia	Prontosan	Granudacyn	Actimaris	Octenilin	Betaseptic
*S. aureus*	1250	1	0	0	0	0	0	0
*S. aureus*	2500	1	0	0	0	0	0	0
*S. aureus*	2000	1	0	0	1	0	0	0
*S. epidermidis*	1250	1	0	0	0	0	0	0
*S. aureus*	1250	1	0	0	1	0	0	0
*E. cloacae*	1250	1	0	0	0	0	0	0
Total eradication		0%	100%	100%	66%	100%	100%	100%

## Data Availability

The data presented in this study are available in the Appendix A.

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
