# Peer review of "Antimicrobial Efficacy of Five Wound Irrigation Solutions in the Biofilm Microenvironment In Vitro and Ex Vivo"

_antibiotics, 2025, doi:10.3390/antibiotics14010025_

Round 1
Reviewer 1 Report
Comments and Suggestions for Authors
The manuscript by Honegger et al. evaluates the efficacy of advanced wound irrigation solutions in reducing bacterial biofilms associated with periprosthetic joint infections (PJIs). Using clinical bacterial strains and abscess communities, five commercial solutions and a skin antiseptic were tested. Octenilin® showed the highest biofilm reduction (3.07 log10), while ActiMaris® excelled against abscess communities (8.27 log10). All but Granudacyn® eradicated bacteria in ex vivo clinical sonication fluids. The findings underscore the potential of these solutions in PJI treatment and the importance of considering bacterial species, biofilm states, and solution composition for optimized therapies.
Overall, this manuscript demonstrates the variable efficacy of advanced wound irrigation solutions in reducing bacterial biofilms and highlights their potential to improve PJI treatment outcomes during revision surgery. However, the authors do not provide sufficient details in the background regarding the significance of their findings. The results are not thoroughly explained, and the result part is poorly written. Taking these points and the comments below into consideration, we recommend the manuscript for publication in your journal following major revisions:
1) The authors investigated the antimicrobial efficacy of the irrigation solutions on biofilms. In Figure 1, Octenilin appears to be the only irrigation solution that shows a statistically significant reduction of S. aureus. While the authors discussed the results in the content, it would be beneficial if they could provide more explanation on the potential reasons why Octenilin is the only effective solution.
2) In Figure 1, Betaseptic demonstrates significant effectiveness compared to the other solutions. Could the authors provide more background information on the components of Betaseptic and clarify the differences between these irrigation solutions? Given its predominant effectiveness, what is the rationale or significance of investigating other products?
3) The authors assessed and compared the efficacy of reducing the bacterial load using the SACs model. Could they provide more background information on why this model was chosen and explain the importance of SACs in this test? Are SAC infections commonly observed in PJIs? Have the authors tested any other models for this evaluation?
4) The third part of the paper focuses on the PJI sonication solution ex vivo assay, but this section is poorly written. The experimental process is unclear, and the results are not adequately explained. Furthermore, the inclusion of the sentence, “All figures and tables should be cited in the main text as Figure 1, Table 1, etc.,” is irrelevant to the main context. This reflects a careless approach in this section of the manuscript. The experiments and results discussed in this paragraph need to be completely rewritten if the editor decides to publish the manuscript.
5) The authors concluded that efficacy depends on (I) the bacterium, (II) bacterial growth phenotype, and (III) location. However, this conclusion is not well-supported by the paper. Figure 1 compares efficacy across different bacteria, but the results are similar among them. Regarding (II) bacterial growth phenotype, the authors only assessed one SACs model, and its significance for bacterial growth phenotypes remains unclear. For (III) location, no experiments and results investigating this aspect are showed in the article. The authors need to provide additional results and explanations to justify this conclusion.
Reviewer 2 Report
Comments and Suggestions for Authors
Format. Please adhere to the Antibiotics Guidelines. Some format must be revised.
1. Microbial names, such as Staphylococcus aureus, Staphylococcus epidermidis, Escherichia coli, and Cutibacterium acnes, are not consistently italicized throughout the manuscript, which violates standard scientific conventions.
2. Subsections lack numbering, making it harder for readers to navigate the manuscript. Add subsection number in the materials and method section.
3. Check the references format. Authors must follow the journal's format. Revision is needed.
Major concern:
1. The introduction does not explain why the specific irrigation solutions were chosen for the study. Were they selected based on prior evidence, widespread clinical use, or novel properties? This should be clarified to provide context. Summarize also standard practices for PJI management, particularly irrigation solutions, to establish a baseline.
2. Provide a brief explanation of biofilm matrix components and how these contribute to resistance against irrigation solutions.
3. While the introduction mentions that irrigation solutions aim to reduce bacterial burden during surgery, it does not elaborate on their specific mechanisms of action or advantages over traditional saline solutions.
4. Although the manuscript claims a lack of comparative studies on non-alcohol-based solutions, this point is not well-supported with references. Stronger citations or examples of existing studies focusing on alcohol-based or single-solution studies could highlight the novelty of this work.
5. Please consider adding summary figures or tables that integrate data across species and conditions for a comprehensive comparison.
6. Begin each section of the results interpretation by summarizing the main outcome (e.g., "Among all tested solutions, Octenilin demonstrated the highest efficacy in biofilm reduction across multiple bacterial species."). Use consistent phrasing to compare solutions (e.g., "Octenilin was statistically superior to Prontosan and Granudacyn for reducing biofilms of S. aureus and C. acnes, but not for S. epidermidis.").
7. Explicitly state whether findings are statistically significant and what this means in context. For non-significant results, discuss their implications. Results part must be reformulated.
8. Line 124 - 125 is unclear.
9. Please add conclusion section. The conclusion can be improved by making it more concise, clearly summarizing the key findings, emphasizing clinical implications, and providing actionable insights for future research.
Round 2
Reviewer 1 Report
Comments and Suggestions for Authors
The revised version of the manuscript demonstrates significant improvements over the original and effectively addresses all the provided comments. The authors have expanded on key aspects, such as the experimental design and the interpretation of results, offering a more comprehensive discussion of their findings. These revisions greatly strengthen the overall quality and clarity of the manuscript.
Reviewer 2 Report
Comments and Suggestions for Authors
After revision, the article has been significantly improved and is more suitable for publication.